# Using hyperspectral leaf reflectance to estimate photosynthetic capacity and nitrogen content across eastern cottonwood and hybrid poplar taxa

**Thu Ya Kyaw**[1]*, **Courtney M. Siegert**[1], **Padmanava Dash**[2], **Krishna P. Poudel**[1], **Justin J. Pitts**[1¤], **Heidi J. Renninger**[1]

1 Department of Forestry, Forest and Wildlife Research Center, Mississippi State University, Starkville, Mississippi, United States of America, 2 Department of Geosciences, Mississippi State University, Starkville, Mississippi, United States of America

¤ Current address: Agronomy Department, University of Florida, Gainesville, Florida, United States of America

* tk758@msstate.edu

**Data Availability Statement:** Our data can be accessed from Scholars Junction: Mississippi State University's Institutional Repository at the

## Abstract

Eastern cottonwood (*Populus deltoides* W. Bartram ex Marshall) and hybrid poplars are well-known bioenergy crops. With advances in tree breeding, it is increasingly necessary to find economical ways to identify high-performing *Populus* genotypes that can be planted under different environmental conditions. Photosynthesis and leaf nitrogen content are critical parameters for plant growth, however, measuring them is an expensive and time-consuming process. Instead, these parameters can be quickly estimated from hyperspectral leaf reflectance if robust statistical models can be developed. To this end, we measured photosynthetic capacity parameters (Rubisco-limited carboxylation rate ($V_{cmax}$), electron transport-limited carboxylation rate ($J_{max}$), and triose phosphate utilization-limited carboxylation rate (TPU)), nitrogen per unit leaf area ($N_{area}$), and leaf reflectance of seven taxa and 62 genotypes of *Populus* from two study plantations in Mississippi. For statistical modeling, we used least absolute shrinkage and selection operator (LASSO) and principal component analysis (PCA). Our results showed that the predictive ability of LASSO and PCA models was comparable, except for $N_{area}$ in which LASSO was superior. In terms of model interpretability, LASSO outperformed PCA because the LASSO models needed 2 to 4 spectral reflectance wavelengths to estimate parameters. The LASSO models used reflectance values at 758 and 935 nm for estimating $V_{cmax}$ ($R^2 = 0.51$ and RMSPE = 31%) and $J_{max}$ ($R^2 = 0.54$ and RMSPE = 32%); 687, 746, and 757 nm for estimating TPU ($R^2 = 0.56$ and RMSPE = 31%); and 304, 712, 921, and 1021 nm for estimating $N_{area}$ ($R^2 = 0.29$ and RMSPE = 21%). The PCA model also identified 935 nm as a significant wavelength for estimating $V_{cmax}$ and $J_{max}$. Therefore, our results suggest that hyperspectral leaf reflectance modeling can be used as a cost-effective means for field phenotyping and rapid screening of *Populus* genotypes because of its capacity to estimate these physicochemical parameters.

following DOI: https://doi.org/10.54718/BACR5952.

**Funding:** This work was supported by the National Institute of Food and Agriculture, U.S. Department of Agriculture under award numbers 2018-67020-27934 to HJR and CMS, and 2018-68005-27636 to HJR and CMS, as well as U.S. Department of Agriculture McIntire Stennis Program under accession numbers: MISZ-067050 to HJR, MISZ-032100 to CMS, and MISZ-0621210 to KPP, respectively. The funders had no role in study design, data collection and analysis, decision to publish, or preparation of the manuscript. There was no additional external funding received for this study.

**Competing interests:** The authors have declared that no competing interests exist.

## Introduction

Production of bioenergy through direct combustion [1] and/or the manufacture of liquid biofuels [2] can reduce the consumption of fossil fuels and associated emissions [3]. In the USA, bioenergy contributed about 7.3% to the energy sector in 2019, which could increase to an additional 18–55% by 2050 [4]. To bolster this increasing trend, short rotation woody crops (SRWCs) can be planted as a biomass feedstock for bioenergy production [5, 6]. Currently in the USA, *Populus* feedstock production for bioenergy makes up a small portion of land use, but there is the potential for large-scale plantations for bioenergy in the future because *Populus* is a promising bioenergy crop. Eastern cottonwood (*Populus deltoides* W. Bartram ex Marshall) and hybrid poplars have preferable traits for producing high biomass yield within a short period of time [7–12]. The biomass yield of *Populus* genotypes can reach 7.5 to 15.2 Mg/ha/year [13]. Due to their rapid growth and high biomass productivity, the United States Department of Energy classifies *Populus* as a potential SRWC for bioenergy production [14]. A *Populus* plantation can be managed with a five-year harvesting cycle [10, 15–17], and their excellent coppicing ability [18–20] eliminates the need for replanting after each harvest. Since *Populus* has a wide range of genetic diversity [21, 22], clonal propagation [23, 24] and the use of genetically improved plant material [3, 25] can maximize biomass yield and will accelerate the development of the bioenergy sector [3, 25]. Moreover, *Populus* feedstocks have desirable attributes needed for producing ethanol and other biofuels, such as high cellulose content, moderate lignin and hemicellulose content, and low ash and extractives [26].

Tree breeding has been used to produce many *Populus* genotypes with the aim of maximizing biomass yield [27, 28]. Tree breeding is usually followed by field trials to compare growth performance across developed genotypes. Since field experiments are laborious, costly, and time-consuming [29], identifying superior genotypes for further testing is needed as an initial screening step to optimize field trials. To this end, photosynthetic capacity can be used as a selection criterion for clonal screening because individuals with enhanced photosynthetic rate under field conditions are likely to result in higher biomass yield [30]. To evaluate photosynthetic capacity, photosynthetic $CO_2$ response curves can be used to calculate key photosynthetic capacity parameters, such as (1) maximum carboxylation rate limited by ribulose 1-5-bisphosphate carboxylase/oxygenase (RuBisCO) activity ($V_{cmax}$), (2) electron transport-limited carboxylation rate ($J_{max}$), and (3) triose phosphate utilization-limited carboxylation rate (TPU) [31, 32]. In addition to photosynthetic capacity parameters, nutrient content in leaves is another important parameter for plant growth because limitations of nutrients can decrease photosynthesis. In plants, nitrogen (N) is a limiting macronutrient [33–35]. N deficiency can reduce leaf area [36], leaf durability, and leaf chlorophyll content. Additionally, leaf N content tends to be linearly correlated with photosynthetic capacity [37, 38], and consequently, variations in leaf N content can affect biomass production [39–41].

Photosynthetic capacity parameters and leaf N content provide valuable information about leaf-level processes. However, measuring photosynthetic $CO_2$ response curves in the field takes about 15–20 minutes per leaf. Analogously, measuring leaf N content requires a laboratory assay to generate results. Therefore, these measurement techniques are not cost-effective for high-throughput screening and large-scale field phenotyping. On the other hand, hyperspectral leaf reflectance can potentially be used as an alternative to quickly estimate photosynthetic capacity parameters and leaf N content. Generally, leaf spectra have low reflectance in the visible region mainly due to chlorophyll absorption and high reflectance in the near-infrared region due to internal leaf scattering with little to no absorption [42]. Therefore, there is a physiological relationship between leaf reflectance and photosynthetic capacity parameters [43–45]. For example, according to Qian et al. [44], when $V_{cmax}$ increases, the reflectance at

the red edge position and the green peak decreases. Leaf reflectance can also be correlated with leaf N content because N-containing bonds cause variation in spectral features [46]. Ye et al. [47] found that when leaf N content increased, spectral reflectance decreased, and this relationship was more significantly observed in the green to near infrared regions. Due to these relationships, if robust statistical models can be developed, hyperspectral leaf reflectance can be used as a proxy for rapid assessment of photosynthetic capacity [48–50], as well as non-destructive and real-time monitoring of leaf N content [51, 52]. On the other hand, creating empirical models for hyperspectral data requires sophisticated model development techniques, and model robustness depends on the quality of sample data measured in the field [53].

While existing vegetation indices (e.g., normalized difference vegetation index, photochemical reflectance index, etc.) might be sufficient for estimating physicochemical parameters [43, 54–56], many studies have also developed more robust models using multiple regression methods to improve the prediction capacity for certain plant species [50, 57, 58]. Hyperspectral data have very narrow bandwidths with tremendous amounts of redundant information leading to potential issues with multicollinearity and high-dimensionality. In this respect, statistical methods, such as stepwise regression, shrinkage (e.g., ridge regression and least absolute shrinkage and selection operator (LASSO)), and dimension reduction (e.g., principal component analysis (PCA)) may be useful for handling hyperspectral data. Hyperspectral data also contain hundreds of spectral bands, and therefore hundreds of predictor variables. If the number of predictor variables is greater than the sample size, stepwise selection of multiple linear regression likely chooses nuisance variables rather than true variables, and consequently, performs poorly for out-of-sample accuracy [59, 60]. Ridge regression includes all predictor variables in the final model, which is disadvantageous when analyzing data containing a large number of predictor variables [59]. To solve these issues, many studies suggested LASSO [61–63] and PCA [64–66] methods. The PCA method is easy and computationally efficient. Its major drawback is a difficulty in interpreting principal components because they represent linear combinations of the original features, and thus, each component uses the collection of all predictor variables [67]. LASSO can overcome this limitation of PCA. Compared to PCA, LASSO is likely to generate a simpler model because LASSO uses regularization, shrinkage, and variable selection, and the final model includes a smaller set of predictor variables [59]. However, due to being a supervised feature extraction method [68–70], LASSO is more computationally demanding than PCA. Both LASSO and PCA methods have the capacity for analyzing hyperspectral data, but each method has its own strengths and weaknesses [71–74]. Therefore, comparing both methods on the same data can help to elucidate these tradeoffs, determine where the methods provide convergent information, and identify the method with better model performance.

In this study, we established two *Populus* plantations for field trials in upland sites of Mississippi each containing over 100 genotypes of *Populus* and its hybrids. We postulated that utilizing the large variability in leaf physicochemical parameters across different *Populus* genotypes and sites would result in robust models for *Populus* screening programs. Therefore, using leaf-level data from 62 *Populus* genotypes across seven taxa measured at two study plantations, we developed hyperspectral leaf reflectance models to estimate photosynthetic capacity parameters and leaf N content. Our main objectives were: (1) to assess whether hyperspectral leaf reflectance can adequately model photosynthetic capacity parameters and leaf N content of eastern cottonwood and hybrid poplar, (2) to compare the performance of models developed using LASSO and PCA methods, and (3) to identify spectral wavelengths that are the most sensitive to these physicochemical parameters. Successful development of predictive models from hyperspectral data can be applied to clonal screenings, enable high-throughput field phenotyping [50, 75], and thus, reduce the costs of *Populus* bioenergy research. Moreover,

photosynthetic capacity parameters are input parameters in many terrestrial biosphere models because plant productivity depends on photosynthetic processes [76–79]. Therefore, information about these photosynthetic capacity parameters, which may be cost-effectively estimated from hyperspectral reflectance data, can be used as inputs for process-based earth system models to simulate terrestrial carbon fluxes [80–82] for improving our understanding of regional and global carbon cycles.

## Materials and methods

### Field and laboratory analysis

**Study area.**   The study area consisted of two plantations containing over 100 *Populus* genotypes located in Monroe County (88˚ 17' W, 33˚ 51' N) and Pontotoc County (88˚ 59' W, 34˚ 8' N), MS. The plantation at the Monroe site was established in April 2018 at 2.7 × 1.8 m spacing. The average height of trees was 6.2 m at the end of the second growing season in 2019. The Monroe site has moderately well-drained, Prentiss fine sandy loam soils [83] and is surrounded by, and previously used for, row crop agriculture with 0–2 percent slopes. This site has soil carbon, soil nitrogen, organic matter, and dry bulk density of 2.29 kg/m$^2$, 0.21 kg/m$^2$, 1.08%, and 1.60 g/cm$^3$, respectively. Based on climate data from a nearby weather station (USW00093862), the Monroe site has an annual temperature range of 11–23˚C, and mean annual rainfall of 1,397 mm [84].

The plantation at the Pontotoc site was established in April, 2019 at 2.7 × 1.8 m spacing. The average height of trees was 5.3 m at the end of the second growing season in 2020. The Pontotoc site has well-drained, Atwood silt loam soils [83] and is also surrounded by, and previously used for, row crop agriculture with 0–2 percent slopes. This site has soil carbon, soil nitrogen, organic matter, and dry bulk density of 3.61 kg/m$^2$, 0.44 kg/m$^2$, 1.83%, and 1.52 g/cm$^3$, respectively. Based on climate data from the Pontotoc Experimental Station, MS (USC00227111), the Pontotoc site has an annual temperature range of 10–22˚C, and mean annual rainfall of 1,482 mm [84].

**Photosynthetic A/C$_i$ curves.**   Gas exchange measurements were conducted in the field using a LI-COR 6400 XTP portable photosynthesis system (LI-COR Biosciences Inc., Lincoln, NE, USA) with a red/blue light source attached. At the Pontotoc site, leaves were within reach, and measurements were made on live leaves still attached to branches on the mid canopy sunlit side of trees. At the Monroe site, leaves could not be accessed due to the height of trees, and a 7.8 m long pole pruner was used to cut branches from the mid to upper parts of crowns on the sunlit side of trees. The cut branches were immediately submerged in water, and approximately 20 cm of the branch end was recut underwater to minimize embolisms [85, 86]. Then, sample leaves attached to cut branches were measured in a sunlit location outside of the field plot. Fully expanded leaves with minimal disease located near the top of branches were selected for these measurements. Photosynthetic A/C$_i$ response curves (A = net $CO_2$ assimilation and C$_i$ = sub-stomatal $CO_2$ concentration inside the leaf) of the sample leaves were measured by setting ambient $CO_2$ levels at 400 ppm and then, 300, 200, 100, and 50 ppm before being increased to 400, 600, and 800 ppm. During measurements, stomatal conductance was monitored. If low stomatal conductance occurred as a result of stress due to removal from the tree (in the case of the Monroe site), a new branch was collected and measured. Measurements were made between 10:00 AM and 2:00 PM on clear sunny days. For each individual during each measurement period, photosynthetic photo flux density, leaf temperature, and relative humidity inside the leaf chamber were controlled under saturating light conditions at 1500 μmol/m$^2$/s, 30˚C, and 40 to 60%, respectively [85–87]. From the A/C$_i$ curves, we calculated three main photosynthetic capacity parameters–$V_{cmax}$, $J_{max}$, and TPU in a Microsoft

Excel solver program that contained non-linear curve-fitting equations from Sharkey et al. [88], which was based on Farquhar et al. [31]. During calculation, the parameters were scaled to a standardized temperature of 25˚C.

We measured a total of 105 leaves ($n$ = 105). Among them, we measured 57 leaves at the Monroe site in July and early September in 2019 and 48 leaves at the Pontotoc site in July in 2020. In total, about 85% of leaf samples were measured in July and 15% in early September. September measurements were made on previously measured genotypes from July because some spectral response curves from leaves measured in July were more variable than the majority of collected data, and thus, these genotypes were re-measured. At the Monroe site, we measured leaf samples from 21 genotypes and five taxa, including *P. deltoides* × *P. deltoides* (D×D, or eastern cottonwood), *P. deltoides* × *P. maximowiczii* A. Henry (D×M), *P. deltoides* × *P. nigra* L. (D×N), *P. deltoides* × *P. trichocarpa* Torr. & Gray (D×T), and *P. trichocarpa* × *P. deltoides* (T×D) (S1 Table). At the Pontotoc site, we measured leaf samples from 48 genotypes and six taxa, including D×D, D×M, D×N, the three-way hybrid *Populus deltoides* × *Populus nigra* × *Populus maximowiczii* ((D×N)×M), D×T, and T×M (*P. trichocarpa* × *P. maximowiczii*) (S1 Table).

**Hyperspectral leaf reflectance.**   Immediately after gas exchange measurements of a leaf, hyperspectral leaf reflectance was measured on the same leaf so that both measurements were made in close proximity temporally and under the same environmental and light conditions. The spectral measurements were made with a portable, single-beam GER 1500 field spectroradiometer (Spectra Vista Corp., Poughkeepsie, NY, USA). The GER 1500 can detect a spectral range of 286.90 to 1089.62 nm and contains 512 spectral bands. It has full-width at half-maximum of 3.20 nm and a nominal bandwidth of approximately 1.50 nm covering parts of the ultraviolet, visible, and near-infrared wavelength regions of the electromagnetic spectrum. Decimal values of spectral wavelengths were rounded to integer values when reporting results.

To calculate the remote sensing reflectance of leaves, we collected two types of spectral data—radiance of the leaf and irradiance of the sky under sunny, cloud-free conditions. From approximately 40 cm above the leaf, radiances were measured on the leaf adaxial surface. Excluding the midrib, four radiance measurements were taken from various locations on each leaf lamina, and these readings were then averaged to obtain mean radiance for individual leaves. Irradiance of the sky was measured by holding the spectroradiometer and its attached fiber-optic cable with diffuser vertically toward the cloud-free sky. Six irradiance readings were recorded between individual radiance measurements, and these readings were then averaged to obtain mean irradiance of the sky. To calculate the reflectance of leaves, mean sky irradiance measured in close temporal proximity to individual radiance readings was chosen. The remote sensing reflectance for each individual leaf was then calculated by dividing mean radiance of the leaf with mean irradiance of the sky.

**Leaf N content.**   Leaves measured for both photosynthetic capacity parameters and hyperspectral reflectance were placed in sealed plastic bags, kept inside a cooler, and transported to the laboratory for N determination. Fresh leaf area was measured using a LI-3100C Area Meter (LI-COR Biosciences Inc., Lincoln, NE, USA) in the laboratory. Leaves were then oven-dried at 60˚C for at least 72 h until all water was evaporated from samples, and their dry weight was measured. Next, dried leaves were ground to a fine powder that could pass through a 0.25 mm sieve and analyzed for N content via elemental combustion analysis (ECS 4010 CHNOS Elemental Analyzer, Costech Analytical Technologies Inc., Valencia, CA, USA). Standards, duplicates, and blanks were included in the analysis for quality assurance. Since all photosynthetic capacity parameters were calculated on an area basis, we calculated leaf N content on a per unit area basis ($N_{area}$; g N/m$^2$) by using leaf mass per area (LMA; calculated as the ratio of

dried leaf weight and fresh leaf area) as follows:

$$N_{\text{area}} = \frac{\text{leaf N weight}}{\text{leaf sample weight}} \times \text{LMA}$$

**Tree biomass.**    To determine if leaf physicochemical parameters were correlated with woody biomass and productivity, the measured *Populus* trees were harvested to estimate their biomass at the end of the second growing season. The freshly harvested trees were separated into leaf and wood components with fresh wood weights measured in the field and fresh leaf weights measured in the laboratory. A wood sample from each tree was also returned to the laboratory and its fresh weight was recorded. Wood samples were oven-dried at 65°C, and their dry weight was measured and used to calculate the dry mass of the field-weighed wood samples to determine dry woody biomass (kg) for each tree.

## Statistical analysis

**Taxa comparisons and correlations across parameters.**    We made statistical comparisons at the taxa level. We grouped the measured leaf sample data based on taxa. We added T×D genotypes into D×T, and the(D×N)×M genotype into D×N due to their small sample sizes (T×D, $n = 2$ leaf samples, and (D×N)×M, $n = 1$ leaf sample) (S1 Table). Therefore, we had five taxa groups. Since analysis of variance (ANOVA) assumptions, such as normal distribution of model residuals and homogeneity of variance across groups, were not met, a non-parametric Kruskal-Wallis test was performed to compare the differences in photosynthetic capacity parameters ($V_{\text{cmax}}$, $J_{\text{max}}$, and TPU), $N_{\text{area}}$, and tree biomass among the five taxa groups and between the Monroe and Pontotoc sites using R version 4.0.4 [89]. If taxa groups differed significantly, a post-hoc Dunn's Kruskal-Wallis multiple comparison test using the Bonferroni method was then performed to identify significant differences among the groups. To determine the relationship among $V_{\text{cmax}}$, $J_{\text{max}}$, TPU, $N_{\text{area}}$, and tree biomass, we used all measured leaf sample data ($n = 105$) and calculated a Pearson correlation coefficient matrix across these parameters.

**Model development: Least absolute shrinkage and selection operator.**    We used all measured leaf sample data ($n = 105$) to develop models for estimating leaf physicochemical parameters from hyperspectral data across measured *Populus* taxa and genotypes. We followed the least absolute shrinkage and selection operator (LASSO) model development method to create hyperspectral leaf reflectance models for each leaf physicochemical parameter. To address the high-dimensional features of hyperspectral data and to reach a sparse solution, LASSO cross validation [90] (i.e., an L1-regularized regression) was applied for variable selection. Predictor variables were hyperspectral wavelength bands, and the response variables were photosynthetic capacity parameters ($V_{\text{cmax}}$, $J_{\text{max}}$, and TPU) and $N_{\text{area}}$.

All statistical analyses were performed in R version 4.0.4 [89]. Using R package 'glmnet' version 2.0.18 [91], *k*-fold cross validation for 'glmnet' was performed by specifying the number of folds as 10. Deviance, which used mean square error for Gaussian models, was assigned as the loss for cross validation. From the cross validation output, the tuning parameter lambda ($\lambda$) at minimum mean cross-validated error was selected. Once the optimal $\lambda$ was determined, data were fit with a generalized linear model through penalized maximum likelihood using LASSO regularization [91]. LASSO minimized overfitting by shrinking the coefficients of variables that were not related with the response variable to zero and thus, the predictor variables with non-zero coefficients were selected for the linear model. Then, $R^2$, root mean square error (RMSE), and root mean square percentage error (RMSPE; calculated as $(\text{RMSE}/\bar{X}) \times 100\%$) of the selected models were calculated.

For model validation, the data were divided into training and testing datasets using the 'trainControl' function available in R package 'caret' version 6.0–84 [92]. Training and testing data included 70% and 30% of the entire data, respectively. The control parameters for cross validation were defined as 10-fold cross validation with 20 repetitions [93]. Then, the averaged values of repeated 10-fold cross-validated RMSE and $R^2$ of the model were calculated.

**Model development: Principal component analysis.** We used all measured leaf sample data ($n = 105$) for principal component analysis (PCA) model development. Using the 'prcomp' function in R, we conducted PCA to create hyperspectral leaf reflectance models for each physicochemical parameter. An unrotated P-mode PCA was conducted on the data matrix (spectral bands × leaves) to produce orthogonal principal components (PCs), of which those with eigenvalues greater than 1.0 were retained [94]. Two PCs were retained as they explained 98.2% of the variance (PC1 = 85.8% and PC2 = 12.4%). Next, linear regression was performed using PC1 and PC2 as predictor variables for each of the photosynthetic capacity parameters and $N_{area}$. Then, the performance of these linear models was evaluated with training and testing data using the methods described above for repeated 10-fold cross validation.

## Results

### Photosynthetic parameters, $N_{area}$, biomass, and leaf reflectance

$V_{cmax}$ ranged from 67.8 to 368.8 with a mean of 184.4 ± 15.7 μmol/m$^2$/s; $J_{max}$ ranged from 71.5 to 402.4 with a mean of 193.5 ± 17.3 μmol/m$^2$/s; TPU ranged from 5.3 to 25.8 with a mean of 13.2 ± 1.2 μmol/m$^2$/s; $N_{area}$ ranged from 1.1 to 4.7 with a mean of 2.5 ± 0.1 g/m$^2$; and age two, whole-tree dry woody biomass ranged from 779.4 to 7288.4 with a mean of 4001.9 ± 341.7 g. These were the results calculated from all measured leaf sample data ($n = 105$).

Among the five taxa groups, D×N had 31% lower $V_{cmax}$ than D×T; D×D had 32% lower $J_{max}$ than D×T; and D×D had 31% lower TPU than D×T. For tree biomass, D×N and T×M were 39% and 54% lower than D×T, and 40% and 54% lower than D×D, respectively (Fig 1). There was no difference in $N_{area}$ among the taxa (Fig 1). Generally, T×M had lower leaf reflectance especially when compared with D×N and D×T (Fig 2). The positions of peak and trough leaf reflectance were mainly observed within the range of 700–800 nm and 900–1000 nm, respectively (Fig 2 and S1 Fig).

Trees at the Monroe site had higher photosynthetic capacity and leaf N content than those at the Pontotoc site. Trees at the Monroe site had 37% higher $V_{cmax}$, 44% higher $J_{max}$, 45% higher TPU, 12% higher $N_{area}$, and 34% higher tree biomass than those at the Pontotoc site (Fig 1). Tree biomass was positively correlated ($P$-value < 0.05) with $V_{cmax}$, $J_{max}$, and TPU, but not with $N_{area}$ ($P$-value = 0.22). $N_{area}$ was positively correlated with $V_{cmax}$, $J_{max}$, and TPU ($P$-value < 0.05) (Table 1).

### Variable selection and model performance

The LASSO model selected the same two wavelengths, 758 and 935 nm to estimate both $V_{cmax}$ and $J_{max}$ (Table 2 and S1 Fig). The TPU model selected three wavelengths, 687, 746, and 757 nm; and the $N_{area}$ model selected four wavelengths, 304, 712, 921, and 1021 nm (Table 2 and S1 Fig). The LASSO model for $N_{area}$ had the lowest RMSPE (21%), followed by TPU (31%), $V_{cmax}$ (31%), and $J_{max}$ (32%). In terms of $R^2$, the LASSO model for TPU was the highest ($R^2$ = 0.56), followed by $J_{max}$ ($R^2$ = 0.54), $V_{cmax}$ ($R^2$ = 0.51), and $N_{area}$ ($R^2$ = 0.29) (Table 2 and Fig 3).

In the PCA method, PC loading in each spectral band indicated that PC1 did not have much variability, however, distinguishability mainly occurred in PC2 (Fig 4). For PC2, the wavelengths that had the highest positive loadings were 429, 431, 437, 439, and 414 nm, and those that had the highest negative loadings were 931, 928, 930, 934, and 927 nm. Both PC1

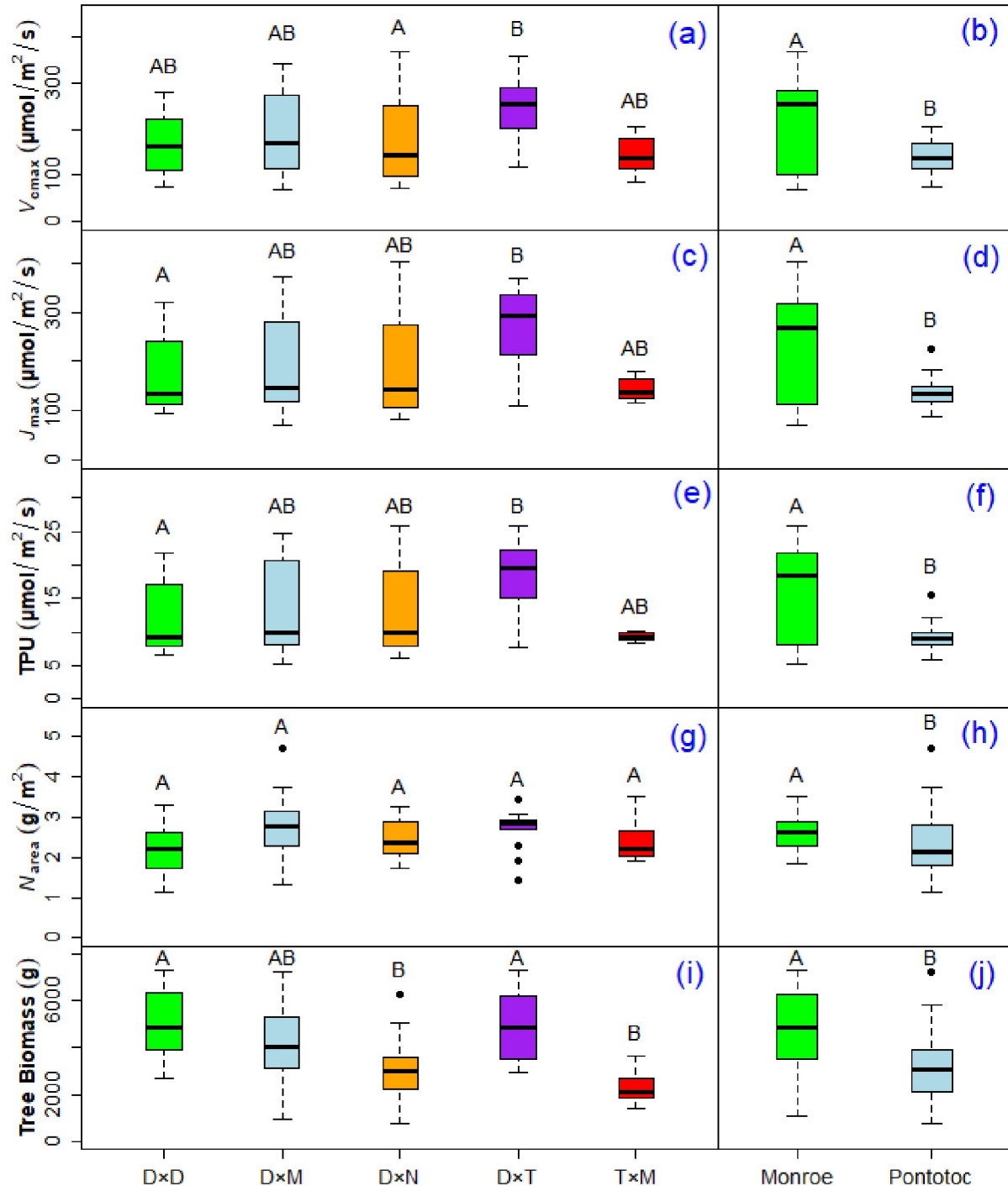

**Fig 1. Box plots showing the variability among five different *Populus* taxa (left panels) and between the Monroe and Pontotoc sites (right panels).** Bars with different letters indicate statistically significant differences ($P$-value < 0.05). Parameters include maximum carboxylation rate of Rubisco ($V_{cmax}$) by (a) taxa and (b) site, maximum electron transport limited carboxylation rate ($J_{max}$) by (c) taxa and (d) site, triose phosphate utilization limited carboxylation rate (TPU) by (e) taxa and (f) site, nitrogen per unit leaf area ($N_{area}$) by (g) taxa, (h) site, and tree biomass by (i) taxa and (j) site. D×T and T×D taxa were combined and shown as D×T, and (D×N)×M was included in D×N (S1 Table). D×D = *P. deltoides* × *P. deltoides*, D×M = *P. deltoides* × *P. maximowiczii*, D×N = *P. deltoides* × *P. nigra*, D×N×M = *P. deltoides* × *P. nigra* × *P. maximowiczii*, D×T = *P. deltoides* × *P. trichocarpa*, T×D = *P. trichocarpa* × *P. deltoides*, and T×M = *P. trichocarpa* × *P. maximowiczii*.

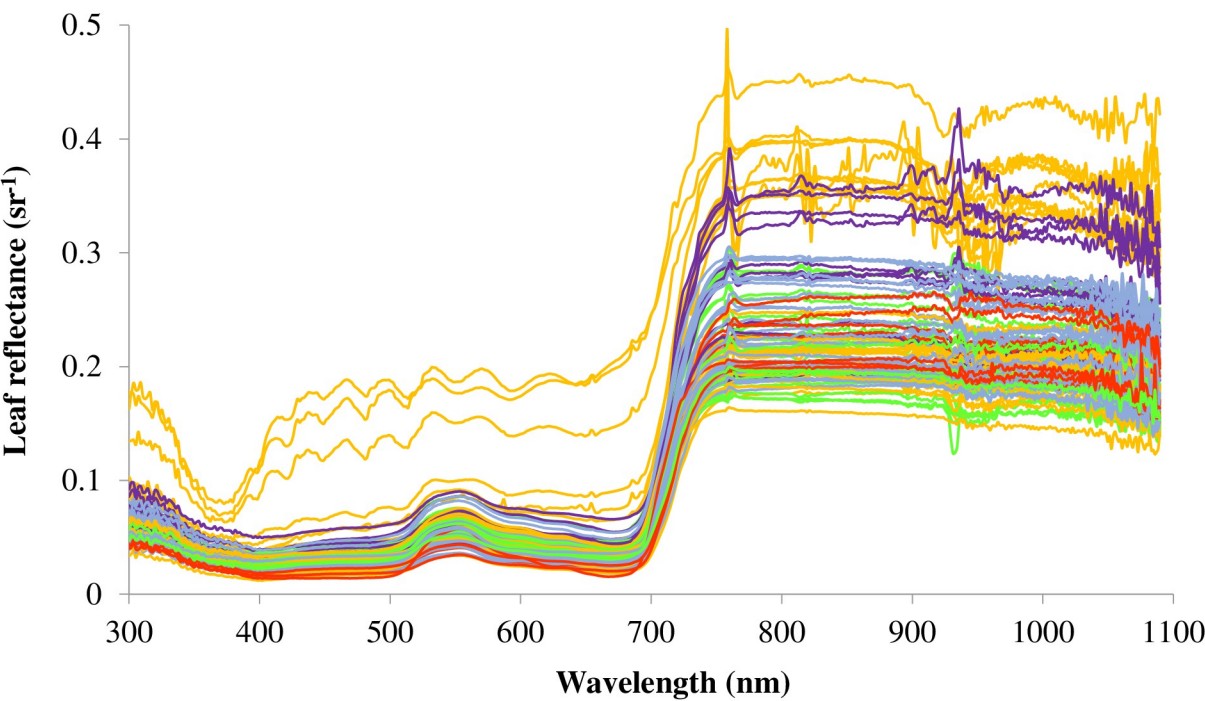

**Fig 2. Leaf reflectance curves of five different *Populus* taxa.** D×T and T×D taxa were combined and shown as D×T, and taxa (D×N)×M was included in D×N (S1 Table). D×D = *P. deltoides* × *P. deltoides*, D×M = *P. deltoides* × *P. maximowiczii*, D×N = *P. deltoides* × *P. nigra*, (D×N)×M = *P. deltoides* × *P. nigra* × *P. maximowiczii*, D×T = *P. deltoides* × *P. trichocarpa*, T×D = *P. trichocarpa* × *P. deltoides*, and T×M = *P. trichocarpa* × *P. maximowiczii*.

and PC2 loadings indicated that the wavelengths near 935 nm were significant (Fig 4), and the same wavelength was also selected by the LASSO models for estimating $V_{\text{cmax}}$ and $J_{\text{max}}$ (Table 2).

The PCA models had comparable RMSPE and $R^2$ as the LASSO models for $V_{\text{cmax}}$, $J_{\text{max}}$, and TPU. Only the PCA model for $N_{\text{area}}$ had substantially lower $R^2$ than the LASSO model for $N_{\text{area}}$ (PCA = 0.15 vs. LASSO = 0.29) but RMSPE was comparable (PCA = 23% vs. LASSO = 21%) (Table 2). Among the PCA models, the PCA model that estimated $N_{\text{area}}$ had the lowest RMSPE (RMSPE = 24%), followed by $V_{\text{cmax}}$ (RMSPE = 31%), $J_{\text{max}}$ (RMSPE = 32%), and TPU (RMSPE = 33%) (Table 2 and Fig 3). The PCA model for $J_{\text{max}}$ had the highest $R^2$ ($R^2$ = 0.53), followed by models for $V_{\text{cmax}}$ ($R^2$ = 0.51), TPU ($R^2$ = 0.50), and $N_{\text{area}}$ ($R^2$ = 0.15).

The model validation using repeated 10-fold cross validation indicated that there was a very slight difference in RMSEs among the training data, test data, and entire dataset for both

**Table 1. Pearson correlation coefficient ($r$) matrix among tree biomass, $V_{\text{cmax}}$, $J_{\text{max}}$, TPU, and $N_{\text{area}}$.**

|  | Tree biomass | $V_{\text{cmax}}$ | $J_{\text{max}}$ | TPU | $N_{\text{area}}$ |
|---|---|---|---|---|---|
| Tree biomass | 1.00 |  |  |  |  |
| $V_{\text{cmax}}$ | **0.33** | 1.00 |  |  |  |
| $J_{\text{max}}$ | **0.35** | **0.97** | 1.00 |  |  |
| TPU | **0.36** | **0.96** | **0.99** | 1.00 |  |
| $N_{\text{area}}$ | 0.12 [ns] | **0.39** | **0.38** | **0.36** | 1.00 |

Non-significant correlations are denoted by "ns" superscript, and the rest with "bold" font are significant at $P$-value $<$ 0.05.

**Table 2. Selected predictor variables in the models and averaged 10-fold cross validation results.**

| Model | Model form | $R^2$ | RMSPE | RMSE (Entire data) | Cross validated RMSE | |
| --- | --- | --- | --- | --- | --- | --- |
| | | | | | Training data | Test data |
| LASSO: $V_{cmax}$ | -30.2 + (601.1 × ρ758.29) + (302.7 × ρ935.71) | 0.51 | 31% | 57.20 | 59.07 | 57.37 |
| LASSO: $J_{max}$ | -44.6 + (796.2 × ρ758.29) + (200.57 × ρ935.71) | 0.54 | 32% | 60.98 | 61.68 | 60.64 |
| LASSO: TPU | -6.4 + (-62.0 × ρ687.03) + (35.2 × ρ745.99) + (59.9 × ρ756.76) | 0.56 | 31% | 4.02 | 4.02 | 4.37 |
| LASSO: $N_{area}$ | 2.0 + (28.3 × ρ303.51) + (-24.4 × ρ711.97) + (0.7 × ρ920.68) + (5.9 × ρ1021.44) | 0.29 | 21% | 0.51 | 0.53 | 0.51 |
| PCA: $V_{cmax}$ | 184.4 + (-1.9 × PC1) + (-5.6 × PC2) + (-0.03 × PC1 × PC2) | 0.52 | 31% | 56.99 | 56.68 | 59.54 |
| PCA: $J_{max}$ | 193.5 + (-2.4 × PC1) + (-5.3 × PC2) + (-0.01 × PC1 × PC2) | 0.53 | 32% | 61.92 | 62.46 | 64.72 |
| PCA: TPU | 13.2 + (-0.2 × PC1) + (-0.3 × PC2) + (-0.0007 × PC1 × PC2) | 0.50 | 33% | 4.30 | 4.20 | 4.65 |
| PCA: $N_{area}$ | 2.5 + (0.0008 × PC1) + (-0.04 × PC2) + (-0.0005 × PC1 × PC2) | 0.15 | 23% | 0.56 | 0.60 | 0.55 |

Training and testing data contained 70% and 30% of the entire data, respectively. ρ is spectral reflectance at the corresponding wavelengths.

LASSO and PCA models (Table 2). Therefore, both LASSO and PCA models were not over-fit. Cross-validation also indicated that RMSEs were generally higher, albeit modestly, for the PCA models compared with the LASSO models.

## Discussion

### Model performance

Our results showed promise in estimating photosynthetic capacity parameters ($V_{cmax}$, $J_{max}$, and TPU) using hyperspectral leaf reflectance (Table 2). In both LASSO and PCA, the models for estimating $N_{area}$ had the lowest $R^2$, but RMSPE was smaller for $N_{area}$ than that of the estimates of $V_{cmax}$, $J_{max}$, and TPU (Table 2). The $N_{area}$ model likely had the smallest RMSPE

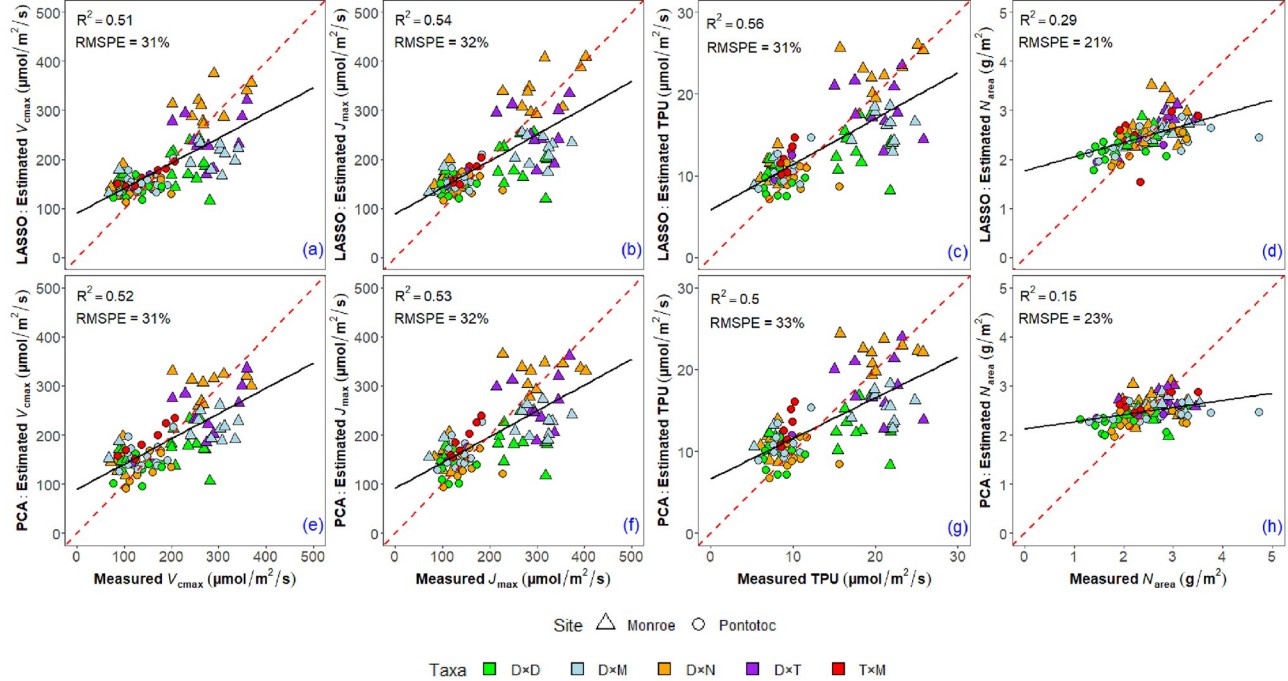

**Fig 3. Scatterplots showing the linear relationship between measured and estimated $V_{cmax}$, $J_{max}$, TPU and $N_{area}$ for the LASSO models: (a), (b), (c), and (d) and the PCA models: (e), (f), (g), and (h).** The solid black line is the regression line, and the dotted red line is the 1:1 line.

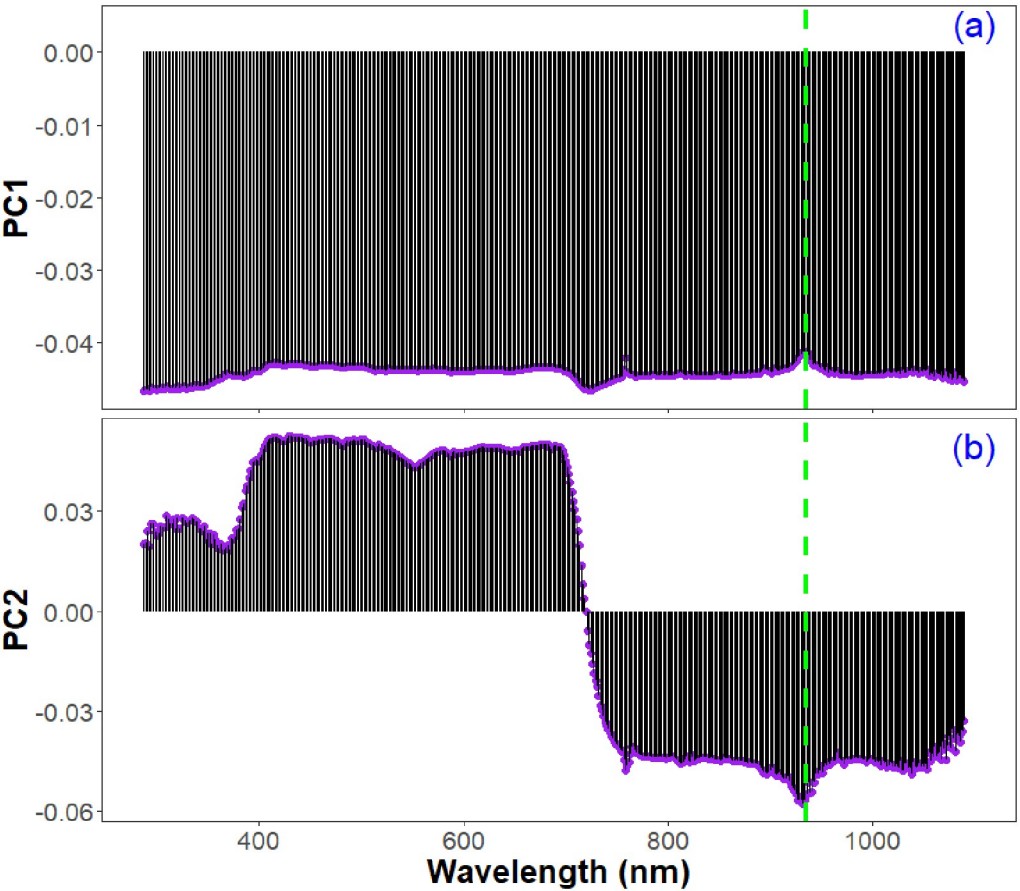

**Fig 4. Lollipop plot showing the loadings of all spectral wavelengths in (a) PC1 and (b) PC2.** As shown by vertical green dashed line, a significant peak and trough occurred at or near 935 nm, which was also selected by the LASSO models for estimating $V_{cmax}$ and $J_{max}$.

because of having smaller variance compared to that of $V_{cmax}$, $J_{max}$, and TPU (Fig 1). The $N_{area}$ model had the smallest $R^2$ probably because $N_{area}$ was weakly correlated with photosynthetic capacity parameters (Table 1) suggesting that a large portion of N in leaves was not involved in photosynthesis, and consequently, might not generate a strong spectral signal. In addition to $N_{area}$, we also developed a model for leaf N content per unit mass (i.e., $N_{mass}$ (%); calculated by dividing leaf N mass with leaf sample mass and then, multiplying with 100). Nonetheless, the $N_{area}$ model ($R^2 = 0.29$ for LASSO and 0.15 for PCA) fit the data better than that of $N_{mass}$ (%) ($R^2 = 0.07$ for LASSO and 0.02 for PCA). $N_{area}$ resulted in a better model likely because $N_{area}$ incorporated the information of both $N_{mass}$ and leaf mass per area in its calculation.

For screening, new genotypes are developed by breeding within and potentially between taxa. Therefore, creating models from taxa- or genotype-specific data may not be robust enough because new genotypes that need to be screened for productivity potential are being developed. In this study, we used all leaf sample data ($n = 105$) measured across 7 taxa and 62 genotypes while developing LASSO and PCA models, with many of these genotypes commonly selected for establishing short rotation plantations across different geographic regions. Our measurements at both study plantations represented a single growing season, but we measured a wide variety of taxa and genotypes to generate large variability in physicochemical

parameters ($V_{cmax}$, $J_{max}$, TPU, and $N_{area}$) and to develop robust models for predicting *Populus* parameters. Therefore, our hyperspectral leaf reflectance captured variability in physiological functioning across a diverse range of *Populus* taxa and genotypes.

LASSO and PCA model development techniques had similar capacity in estimating $V_{cmax}$, $J_{max}$, and TPU. However, the LASSO models used only 2 to 4 wavelengths to estimate all response variables, while the PCA models used 2 PCs, including all 512 wavelengths. Consequently, it is easier to interpret the results of the LASSO models compared to the PCA models. On the other hand, the LASSO method requires more sophisticated analysis. Since LASSO is a supervised feature extraction method, it needs to define an optimal value for the tuning parameter ($\lambda$) using cross-validation procedures to adjust the balance between sparsity (i.e., the model containing few non-zero coefficients) and model prediction accuracy [59, 71]. Furthermore, LASSO selects only one variable from the group if predictor variables within a group are highly correlated [67, 95]. Despite these setbacks, our findings suggested that sparse statistical modeling with LASSO could simplify models by selecting a small subset of predictor variables, which was especially helpful for analyzing data with high dimensionality and complexity in which small and critical differences in data could not be easily detected [72]. Therefore, we found that the LASSO method was able to develop acceptable models for the hyperspectral data in which the number of predictor variables was significantly larger than the sample size.

In contrast to the LASSO method, the PCA method was an unsupervised dimension reduction technique. Consequently, it is easier and quicker to generate results with minimum input from the analyst. The PCA method transforms the high-dimensional space of correlated variables into a new feature space of high variance, and consequently can achieve the tasks of both feature and noise reduction [96]. However, the LASSO method does not transform the data, and consequently, the data might still contain noise [97]. The disadvantage of the PCA method is that, while constructing the principal components to minimize the correlations among them, the principal components do not necessarily tie in with the interest of specific model outcome [67]. Because all the variables exist in each principal component, it is harder to interpret the direct relationship between individual spectral wavelengths and leaf physicochemical parameters. The model performance between LASSO and PCA methods was equivalent in estimating photosynthetic capacity parameters, however, the LASSO model outperformed the PCA model in estimating $N_{area}$ (Table 2). Therefore, the LASSO method was able to provide a simple and robust model for all response variables even though it required more time and effort during model development.

We also analyzed previously published vegetation indices in linear regressions by using our measured data in order to evaluate whether these vegetation indices were comparable with our LASSO and PCA models in estimating $V_{cmax}$, $J_{max}$, TPU, and $N_{area}$. We calculated $R^2$ and RMSE of vegetation indices and compared these parameters with our best-fit LASSO and PCA models. We found that published vegetation indices did not perform better than our models because our models, particularly the LASSO models, had the highest $R^2$ and the lowest RMSE for estimating photosynthetic capacity parameters and $N_{area}$ in our *Populus* trees (Table 3). This was likely because these vegetation indices were developed using other species and to estimate other physiological and leaf- and canopy-level parameters than we estimated. After LASSO and PCA models, the hyperspectral vegetation indices developed for estimating canopy-level growth parameters in cotton (*Gossypium hirsutum* L.) and leaf-level chlorophyll content in a wide range of plants [98–102] showed the next best performance for estimating $V_{cmax}$, $J_{max}$, and TPU (Table 3 and S2 Table). Regarding $N_{area}$, the LASSO model performed the best. However, hyperspectral red edge index ($\rho 738/\rho 720$) [103] and linear red edge index [98] previously used for estimating leaf area index in wheat and canopy N in cotton, and

**Table 3. Comparison of model performance among the LASSO model, the PCA model, and published vegetation indices.**

| Source | Formula | $V_{cmax}$ | | $J_{max}$ | | TPU | | $N_{area}$ | |
|---|---|---|---|---|---|---|---|---|---|
| | | $R^2$ | RMSE | $R^2$ | RMSE | $R^2$ | RMSE | $R^2$ | RMSE |
| LASSO | This study | 0.51 | **57.20** | 0.54 | **60.98** | 0.56 | **4.02** | 0.29 | **0.51** |
| PCA | This study | 0.52 | **56.99** | 0.53 | **61.92** | 0.50 | **4.30** | 0.15 | 0.56 |
| Carter et al. [104] | ρ695/ρ420 | 0.02[ns] | 81.21 | 0.03[ns] | 88.67 | 0.03[ns] | 5.97 | 0.01[ns] | 0.60 |
| Carter et al. [104] | ρ695/ρ760 | 0.00[ns] | 81.85 | 0.00[ns] | 89.88 | 0.00[ns] | 6.05 | 0.03[ns] | 0.60 |
| Datt [99] | ρ672/(ρ550×ρ708) | 0.08 | **78.62** | 0.09 | **85.74** | 0.11 | **5.72** | 0.03[ns] | 0.60 |
| Gamon (Photochemical Reflectance Index) [105] | (ρ531-ρ570)/(ρ531+ρ570) | 0.00[ns] | 81.80 | 0.00[ns] | 89.70 | 0.01[ns] | 6.04 | 0.04[ns] | 0.60 |
| Gitelson and Merzlyak [106] | ρ750/ρ705 | 0.05 | 79.80 | 0.04 | 88.22 | 0.03[ns] | 5.97 | 0.15 | 0.56 |
| Gitelson and Merzlyak [107] | (ρ750-ρ705)/ (ρ750+ρ705) | 0.03[ns] | 80.53 | 0.02[ns] | 88.88 | 0.02[ns] | 6.00 | 0.13 | 0.57 |
| Gitelson and Merzlyak [108] | ρ750/ρ550 | 0.06 | 79.28 | 0.04 | 87.94 | 0.03[ns] | 5.96 | 0.11 | 0.57 |
| Gitelson and Merzlyak [108] | ρ750/ρ700 | 0.02[ns] | 80.88 | 0.01[ns] | 89.31 | 0.01[ns] | 6.03 | 0.09 | 0.58 |
| Gitelson et al. (Green Normalized Difference Vegetation Index) [109] | (ρ750-ρ550)/(ρ750+ρ550) | 0.04 | 80.23 | 0.03[ns] | 88.76 | 0.02[ns] | 6.00 | 0.10 | 0.58 |
| Gitelson et al. [100] | 1/ρ700 | 0.16 | **74.85** | 0.20 | **80.34** | 0.21 | **5.40** | 0.00[ns] | 0.61 |
| Gupta et al. (Hyperspectral Red Edge Index) [103] | ρ735/ρ720 | 0.06 | 79.54 | 0.05 | 87.59 | 0.04 | 5.93 | 0.17 | 0.55 |
| Gupta et al. (Hyperspectral Red Edge Index) [103] | ρ741/ρ717 | 0.01[ns] | 81.26 | 0.01[ns] | 89.48 | 0.01[ns] | 6.03 | 0.16 | 0.56 |
| Gupta et al. (Hyperspectral Red Edge Index) [103] | ρ747/ρ708 | 0.06 | 79.49 | 0.05 | 87.84 | 0.04[ns] | 5.95 | 0.17 | 0.55 |
| Gupta et al. (Hyperspectral Red Edge Index) [103] | ρ738/ρ720 | 0.07 | 78.78 | 0.07 | 86.75 | 0.06 | 5.88 | 0.19 | 0.55 |
| Le Maire et al. [101] | (ρ749-ρ720)-(ρ701-ρ672) | 0.29 | **69.06** | 0.30 | **75.48** | 0.27 | **5.18** | 0.21 | **0.54** |
| Maccioni et al. [102] | (ρ780-ρ710)/(ρ780-ρ680) | 0.12 | **76.57** | 0.12 | **84.25** | 0.10 | **5.73** | 0.25 | **0.52** |
| Penuelas et al. (Structure Insensitive Pigment Index) [110] | (ρ800-ρ445)/(ρ800-ρ680) | 0.03[ns] | 80.47 | 0.04 | 88.09 | 0.04 | 5.94 | 0.00[ns] | 0.61 |
| Raper and Varco (Linear Red Edge Index) [98] | 700+40[(ρ670+ρ780)/2-ρ700]/ (ρ740-ρ700) | 0.12 | **76.95** | 0.11 | **84.92** | 0.09 | **5.78** | 0.24 | **0.53** |
| Sims and Gamon [111] | (ρ750-ρ445)/(ρ705-ρ445) | 0.06 | 79.26 | 0.06 | 87.36 | 0.04 | 5.92 | 0.17 | 0.55 |
| Sims and Gamon [111] | (ρ800−680)/(ρ800+ρ680−2ρ445) | 0.03[ns] | 80.64 | 0.04[ns] | 88.29 | 0.03[ns] | 5.95 | 0.00[ns] | 0.61 |
| Sims and Gamon [111] | (ρ800-ρ445)/(ρ680-ρ445) | 0.01[ns] | 81.36 | 0.01[ns] | 89.38 | 0.02[ns] | 6.00 | 0.03[ns] | 0.60 |
| Stimson et al. (Hyperspectral Normalized Difference Vegetation Index) [112] | (ρ860-ρ690) /(ρ860 +ρ690) | 0.00[ns] | 81.75 | 0.01[ns] | 89.56 | 0.01[ns] | 6.03 | 0.02[ns] | 0.60 |
| Vogelmann et al. [113] | ρ740/ρ720 | 0.07 | 78.85 | 0.07 | 86.82 | 0.06 | 5.88 | 0.19 | 0.55 |
| Wen et al. [114] | ρ764/ρ716 | 0.00[ns] | 81.71 | 0.00[ns] | 89.88 | 0.00[ns] | 6.06 | 0.15 | 0.56 |
| Zarco-Tejada et al. (Red Edge Optical Index) [115] | ρ750/ρ710 | 0.06 | 79.28 | 0.05 | 87.63 | 0.04 | 5.93 | 0.18 | 0.55 |

Spectral reflectance is denoted by ρ, and non-significant correlations are indicated by the "ns" superscript. The models and vegetation indices having the lowest RMSE for estimating each parameter are bolded. The sources of the alphabetically ordered vegetation indices and their study objectives are described in S2 Table.

vegetation indices developed by Maccioni et al. [102] and Le Maire et al. [101] for estimating leaf chlorophyll content in trees outperformed the PCA model (Table 3 and S2 Table).

## Interpretation of selected wavelengths

As with previous studies [32, 116], we found a strong positive correlation between $V_{cmax}$ and $J_{max}$ ($r = 0.97$) (Table 1) likely due to their coordinated processes in $CO_2$ assimilation. The LASSO models selected the same two wavelengths: 758 and 935 nm for estimating both $V_{cmax}$ and $J_{max}$ (Table 2 and S1 Fig). PCA also indicated 935 nm as a significant wavelength in both PC1 and PC2 (Fig 4). In our leaf reflectance curves (Fig 2 and S1 Fig), peaks and troughs of spectral reflectance mainly occurred at 758 nm and 935 nm. The wavelength 758 nm is an important wavelength for photosynthesis as it corresponds with the sun-stimulated fluorescence emission spectrum in plants [117, 118]. The spike in reflectance near 758 nm (Fig 2 and

S1 Fig) is the spectral signature caused by chlorophyll molecules that re-emit light to dissipate energy and return from excited to non-excited states [118]. In addition, chlorophyll fluorescence emission can be a sign of plant water stress, and steady-state chlorophyll fluorescence emission can reduce net $CO_2$ assimilation [119–122]. According to Frankenberg et al. [123], Fraunhofer lines at 758.8 and 770.1 nm can be used for retrieving chlorophyll fluorescence. Satellite-derived solar-induced chlorophyll fluorescence data have also been suggested as a proxy for estimating and mapping $V_{cmax}$ for global climate change studies [124–126]. Moreover, chlorophyll fluorescence emission can be an indicator of increased photorespiration, which can reduce the rate of photosynthetic capacity [127]. The near infrared wavelength 935 nm is commonly identified as a key wavelength in several vegetation indices. For example, this wavelength was found in the normalized difference vegetation index (NDVI = ($\rho$935 –$\rho$661)/ ($\rho$935 + $\rho$661), in which $\rho$ = spectral reflectance) used to evaluate the visual quality and the drought stress responses of turfgrasses [128–130] as well as to explore the photosynthetic responses under varying salinity stresses in seashore paspalum (*Paspalum vaginatum* Swartz) ecotypes [131]. Furthermore, according to Wang et al. [132], 935 nm was sensitive to leaf greenness and chlorophyll content [133]. Moreover, 935 nm was associated with thermal dissipation of absorbed energy under stress conditions. For example, Zhang et al. [134] observed that non-photochemical quenching (i.e., a mechanism used by plants to protect from adverse effects of high light intensity by dissipating the excessively absorbed radiant energy into heat in the photosystem II antenna complexes) was negatively correlated with the spectral reflectance range 935–945 nm under salinity stress. There was also a significant inverse relationship between non-photochemical quenching and light-adapted fluorescence yield [135, 136] because the fluorescence signal could originate from photosystem II [137]. In addition to its correlation with the leaf chlorophyll fluorescence signal under different abiotic stresses [66, 134], reflectance at 935 nm is related with the water content in plant leaves [138] and could provide potential for understanding leaf water status [139].

The TPU model selected remote sensing reflectance at 687, 746, and 757 nm (Table 2 and S1 Fig). Due to its capacity to detect chlorophyll fluorescence signal, the wavelength 757 nm has been widely used for modeling terrestrial gross primary productivity from space [140–145]. In addition, 687 nm is also used for sun-induced fluorescence retrieval for estimating leaf- and canopy-level net photosynthesis of vegetation [117, 146–149]. The absorption maximum positions of the excitation spectra of fluorescence also occurred at 746 nm in pea plants [150]. The wavelength 746 nm is also associated with photosynthetic processes. Traditionally, the wavelength range of 400–700 nm is considered photosynthetically active radiation (PAR). However, recent studies suggested wavelengths 711 nm, 723 nm, and 746 nm as extended PAR wavelengths as well [151, 152] because these far-red wavelengths also manifested comparable efficiency at driving canopy photosynthesis and yield of both $C_3$ and $C_4$ species and their cultivars [152]. The reflectance at 746 nm may distinguish levels of accessary pigments with Sonobe et al. [153] finding that reflectance at 746 nm had a strong negative correlation with carotenoid concentration in shade-grown tea leaves. Therefore, the predictors included in our TPU model (687, 746, and 757 nm) are in agreement with formerly proposed wavelengths for estimating photosynthetic capacity.

Our hyperspectral leaf reflectance model for estimating $N_{area}$ had lower $R^2$ (0.29 for the LASSO model and 0.15 for the PCA model) than models for estimating $V_{cmax}$, $J_{max}$, and TPU (Table 2). Nevertheless, all four wavelengths selected in our $N_{area}$ LASSO model have been reported elsewhere as critical wavelengths, which were largely associated with, and also useful for, estimating leaf N content in different plant species. Our best model for estimating $N_{area}$ contained four wavelengths– 304, 712, 921, and 1021 nm (Table 2 and S1 Fig). The first wavelength, 304 nm, is in the ultraviolet-B (UV-B) region, which extends from 280 to 320 nm

[154–156]. Leaf protective chemicals have been shown to absorb UV-B radiation at 305 nm [157]. Leaves with more protective chemicals to absorb UV-B radiation might have less N content because protective compounds (e.g., flavonoid content) in plants were found to have a negative relationship with leaf N content [158]. The wavelength near 712 nm was also highly correlated with total N content in cotton plants (*Gossypium hirsutum* L.) [159]. In Chinese cabbage (*Brassica campestris* L. ssp. *Pekinensis* 'Norgangbom') leaves, 710 nm was correlated with chlorophyll content and was also a significant wavelength for predicting leaf N [160]. According to Zhao et al. [41], changes in reflectance at 710 nm could detect levels of N deficiency because of its negative relationship with leaf N in corn (*Zea mays* L.). Besides agricultural crops, 711 nm was one of the correlated spectral bands with leaf N in balsam fir (*Abies balsamea* L. Mill) [161]. The wavelength near 921 nm can be a leaf N predictor as well. Zhao et al. [162] found that the reflectance at 920 nm was associated with leaf N concentration and canopy N density in winter wheat (*Triticum aestivum* L.), and the reflectance at this wavelength was controlled by leaf structural features. Moreover, Tarpley et al. [163] used 920 nm for estimating leaf N content in cotton, and Yu et al. [164] used 921 nm for estimating whole-plant total (including leaves, stems, and roots) N content in pepper (*Capsicum annuum* L.) plants. The wavelength near 1021 nm also showed a relationship with leaf N content. Due to the N-H stretch from protein (i.e., the stretching vibrations of the strong molecular bonds between hydrogen atoms and nitrogen atoms), the N absorption feature centers at 1020 nm in leaves [165]. Moreover, 1020 nm was used as a N absorption feature for estimating canopy N in sagebrush (*Artemisia* spp.) [166] and as a protein absorption feature for estimating leaf N concentrations in grass [167]. Therefore, previously documented studies support the wavelengths selected in our LASSO models for estimating $V_{cmax}$, $J_{max}$, TPU, and $N_{area}$.

## Conclusion

LASSO and PCA model development methods showed similar capacity for estimating photosynthetic capacity parameters, but the LASSO model outperformed the PCA model in estimating $N_{area}$. For analyzing hyperspectral data, the LASSO models resulted in superior model interpretability compared to the PCA models. With only a few spectral wavelengths, narrow-band hyperspectral reflectance can estimate photosynthetic capacity and leaf N content of highly variable eastern cottonwood and hybrid poplar genotypes. This is particularly critical when the most important spectral bands are adjacent to each other (within about 10 nm in TPU model; Table 2) suggesting that high spectral resolution is needed for spectral precision and modeling biophysical traits. Previous physiological studies also found that wavelengths selected by our LASSO models were strongly correlated with photosynthetic capacity and leaf N content. The results of our single-leaf studies corroborated that sun-induced chlorophyll fluorescence retrieval bands (687 and 758 nm) [117, 146–149] could predict $V_{cmax}$, $J_{max}$, and TPU. This suggests that sun-induced fluorescence techniques can be successful in estimating $V_{cmax}$ and $J_{max}$ of *Populus* plantations at the ecosystem level. In total, hyperspectral leaf reflectance can be used as a cost-effective means for high-throughput phenotyping and rapid clonal screening due to its ability to model photosynthetic capacity and leaf N content of a wide variety of *Populus* genotypes.

## Supporting information

**S1 Fig. Locations of wavelengths selected by the LASSO models for estimating $V_{cmax}$, $J_{max}$, TPU, and $N_{area}$.** D×T and T×D taxa were combined and shown as D×T, and (D×N)×M taxa was included in D×N (S1 Table). D×D = *P. deltoides* × *P. deltoides*, D×M = *P. deltoides* × *P. maximowiczii*, D×N = *P. deltoides* × *P. nigra*, (D×N)×M = *P. deltoides* × *P. nigra* × *P.*

*maximowiczii*, D×T = *P. deltoides* × *P. trichocarpa*, T×D = *P. trichocarpa* × *P. deltoides*, and T×M = *P. trichocarpa* × *P. maximowiczii*.
(DOCX)

**S1 Table. List of taxa and genotypes measured at the Monroe and Pontotoc sites.** Monroe and Pontotoc columns describe the number of leaves on which measurements were made and measurement dates. There were a total of seven taxa and 62 genotypes measured at both study sites.
(DOCX)

**S2 Table. Study objectives of selected vegetation indices.**
(DOCX)

## Acknowledgments

This publication is a contribution of the Forest and Wildlife Research Center, Mississippi State University. We thank Dr. Randall J. Rousseau for supervising the *Populus* plantations. We also thank two anonymous reviewers for their constructive suggestions.

## Author Contributions

**Conceptualization:** Thu Ya Kyaw, Courtney M. Siegert, Heidi J. Renninger.

**Data curation:** Thu Ya Kyaw, Justin J. Pitts, Heidi J. Renninger.

**Formal analysis:** Thu Ya Kyaw, Courtney M. Siegert.

**Funding acquisition:** Courtney M. Siegert, Heidi J. Renninger.

**Investigation:** Thu Ya Kyaw, Courtney M. Siegert, Heidi J. Renninger.

**Methodology:** Thu Ya Kyaw, Courtney M. Siegert, Padmanava Dash, Krishna P. Poudel, Heidi J. Renninger.

**Project administration:** Thu Ya Kyaw, Courtney M. Siegert, Heidi J. Renninger.

**Resources:** Thu Ya Kyaw, Courtney M. Siegert, Padmanava Dash, Heidi J. Renninger.

**Supervision:** Thu Ya Kyaw, Courtney M. Siegert, Heidi J. Renninger.

**Validation:** Thu Ya Kyaw, Courtney M. Siegert.

**Writing – original draft:** Thu Ya Kyaw.

**Writing – review & editing:** Courtney M. Siegert, Padmanava Dash, Krishna P. Poudel, Justin J. Pitts, Heidi J. Renninger.

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
