## [Decision Letter · Decision Letter 0]

23 Dec 2021

PONE-D-21-35240Using hyperspectral leaf reflectance to estimate the photosynthetic capacity and nitrogen content of eastern cottonwood and hybrid poplar genotypesPLOS ONE

Dear Dr. Kyaw,

Thank you for submitting your manuscript to PLOS ONE. After careful consideration, we feel that it has merit but does not fully meet PLOS ONE’s publication criteria as it currently stands. Therefore, we invite you to submit a revised version of the manuscript that addresses the points raised during the review process.

We look forward to receiving your revised manuscript.

Kind regards,

Janusz J. Zwiazek

Academic Editor

PLOS ONE

Journal Requirements:

( This material is based upon work that is supported by the National Institute of Food and Agriculture, U.S. Department of Agriculture under award numbers 2018-67020-27934 to HJR and CMS, and 2018-68005-27636 to HJR and CMS, as well as U.S. Department of Agriculture McIntire Stennis Program under accession numbers: MISZ-067050 to HJR, MISZ-032100 to CMS, and MISZ-0621210 to KPP. 

The funders had no role in study design, data collection and analysis, decision to publish, or preparation of the manuscript.)

5. Please include your tables as part of your main manuscript and remove the individual files. Please note that supplementary tables (should remain/ be uploaded) as separate "supporting information" files.

Additional Editor Comments:

Both reviewers agree that this is a potentially a valuable contribution. However, they raise a number of important points that need to be addressed by the authors before the manuscript can be recommended for publication.

Reviewers' comments:

Reviewer's Responses to Questions

**Comments to the Author**

1. Is the manuscript technically sound, and do the data support the conclusions?

Reviewer #1: Yes

Reviewer #2: Yes

2. Has the statistical analysis been performed appropriately and rigorously? 

Reviewer #1: Yes

Reviewer #2: Yes

3. Have the authors made all data underlying the findings in their manuscript fully available?

Reviewer #1: Yes

Reviewer #2: Yes

4. Is the manuscript presented in an intelligible fashion and written in standard English?

Reviewer #1: Yes

Reviewer #2: Yes

5. Review Comments to the Author

Reviewer #1: The authors proposed using hyperspectral leaf reflectance to estimate the photosynthetic capacity and nitrogen content, and modeled it by LASSO and PCA, so as to verify that hyperspectral reflectance modeling can be used as a cost-effective means to conduct extensive field phenotyping to quickly screen Populus genotypes. The language is fluent and there are few small mistakes. The methods are innovative, with strong innovation, reliable experimental design and detailed data, however, there are some problems, such as the lack of organization of each part and disorderly writing. The manuscript should be substantial revisions.

Introduction

The second paragraph (Line 65), is necessary to strengthen the importance of "photosynthetic parameters" and "leaf nitrogen content".

The third paragraph (Line 87) needs to introduce the relationship between spectral data and "photosynthetic parameters", "leaf nitrogen content per unit area". For example, how does the spectral data behave when the net photosynthetic rate is high or leaf nitrogen is high? And clarify the advantages and disadvantages of the field experiments and hyperspectral leaf reflectance.

The fourth paragraph (Line 104) puts forward many methods for analyzing hyperspectral data. Why did the author choose LASSO and PCA? To the best of my knowledge, the stepwise regression of multiple linear regression and ridge regression are also good methods to deal with multicollinearity. What are the advantages and disadvantages of the two models selected? These should be described in the article.

The last paragraph (Line 123) of Introduction should add some introduction concerning the importance of poplar in Mississippi, such as planting area, age of forest, and annual yield and so on.

Materials and Methods

Line 164, photosynthetic traits measurement should be described more detailed, measurement time and conditions should be clarified. See Liu et al. (2021).

Liu X, Zhang Q, Song M, Wang N, Fan P, Wu P, Cui K, Zheng P, Du N, Wang H, and Wang R. Physiological response of Robinia pseudoacacia and Quercus acutissima seedlings to repeated drought-rewatering under different planting methods. Frontiers in Plant Science; 2021; 12:760510.

Line 224, why this study used leaf N content on a per unit area instead of leaf N content on a per unit mass?

Discussion

There are too many paragraphs in Discussion, and the meaning of some paragraphs is unclear, for example, what is the significance of the interpretation of the wavelength at 935 nm (Line 423)? It is necessary to sort out the structure again. The paragraphs on Line 452 and 466 are the discussion of leaf nitrogen content, which can be combined.

It is suggested to separate the Discussion in some parts including, (1) the relationship between traits and spectral data, (2) the discussion of LASSO model, (3) the discussion of PCA model, (4) a comparison among the two models and any other models. The parts above are for your reference only, but they are not appropriate as subtitles. The authors may either divide the discussion into several parts according to the content; or merges paragraphs with similar content.

Reviewer #2: The manuscript reports a study that estimated the photosynthetic capacity and nitrogen content of eastern cottonwood and hybrid Poplar genotypes using hyperspectral leaf reflectance. The results help quickly screen Poplar genotypes with maximum yield potential. However, the manuscript has some significant flaws with the method description. A major revision should be conducted before considering for publication in PLOS ONE. Key concerns with the manuscript are summarised below.

1, What are the soil physicochemical properties (i.e., total N content, organic matter) in those study areas, i.e., Monroe County, Pontotoc County；

2, The authors need to provide more detailed information about the gas exchange measurement, i.e., leaf position in the branches, were the selected leaves fully expanded?

3, How many times did the authors determine the photosynthetic parameters in mid-July and early September in 2019 at the Monroe site and mid-July 2020 at the Pontotoc site? As we know, every time the photosynthetic parameters of those 62 genotypes were determined, the measurement should be conducted at a same time or during the same period, in order to make the comparison between different genotypes reasonable, however, the mid-July and early September in 2019 were a long period.

4, How many repetitions of the photosynthetic parameters measurement did the authors conducted for each genotypes? According to the description in the manuscript, the repetitions of the photosynthetic parameters measurement for each genotypes seemed less than 3, which was not sufficient for the statistical analysis.

5, In the Fig.1, Fig.3, Table 1 and Table, how many values for each parameter of each Populus genotype were used for the statistical analyses, i.e., n=?

6, The leaf spectra has been successfully used for representing the leaf N content, but in this research, why the R2 of the LASSO or PCA model for Narea was the lowest compared with that of Vcmax, Jmax and TPU (Table 3)? The value was only 0.29 or 0.15, it seemed that the leaf spectral reflectance showed weak correlation with N conten.

7, The hyperspectral leaf reflectance would be easily influenced by the leaf growth status and its surroundings, were all the parameters determined at the same time and in the same conditions, if not, how did the authors ensure that the leaf reflectance varied just as the photosynthetic parameters changed? And was the establishment of the relationship between these parameters reasonable?

8, The authors intended to establish methods for rapidly estimating those physicochemical parameters of different Populus genotypes, and quickly screening genotypes with maximum yield potential. In this research, there were 62 Populus genotypes, maybe there should be 62 group data of photosynthetic parameters, leaf N content and hyperspectral leaf reflectance values, respectively. However, only 7 group data were showed in Figure 1. Did the authors establish the correlations between these two parameters by using these 7 group data or 62 group data?

6. PLOS authors have the option to publish the peer review history of their article (what does this mean?). If published, this will include your full peer review and any attached files.

Reviewer #1: No

Reviewer #2: No

---

## [Author Response · Author response to Decision Letter 0]

6 Feb 2022

We thank two anonymous reviewers for their helpful comments and constructive suggestions for improving our manuscript for publication. We addressed the questions of each reviewer and made a substantial revision in this revised manuscript. We also thank the academic editor for giving us the opportunity to revise our manuscript.

---

## [Editor Report · Decision Letter 1]

17 Feb 2022

Using hyperspectral leaf reflectance to estimate photosynthetic capacity and nitrogen content across eastern cottonwood and hybrid poplar taxa

PONE-D-21-35240R1

Dear Dr. Kyaw,

We’re pleased to inform you that your manuscript has been judged scientifically suitable for publication and will be formally accepted for publication once it meets all outstanding technical requirements.

Kind regards,

Janusz J. Zwiazek

Academic Editor

PLOS ONE
---

## [Editor Report · Acceptance letter]

1 Mar 2022

PONE-D-21-35240R1 

Using hyperspectral leaf reflectance to estimate photosynthetic capacity and nitrogen content across eastern cottonwood and hybrid poplar taxa 

Dear Dr. Kyaw:

I'm pleased to inform you that your manuscript has been deemed suitable for publication in PLOS ONE. Congratulations! Your manuscript is now with our production department. 

Kind regards, 

on behalf of

Professor Janusz J. Zwiazek 

Academic Editor

PLOS ONE